# Rapid and unconditional parametric reset protocol for tunable superconducting qubits

Yu Zhou [1,4], Zhenxing Zhang[1,4], Zelong Yin[1], Sainan Huai[1], Xiu Gu[1], Xiong Xu[1], Jonathan Allcock[1], Fuming Liu[1], Guanglei Xi[1], Qiaonian Yu[1], Hualiang Zhang[1], Mengyu Zhang[1], Hekang Li [2,3], Xiaohui Song[2,3], Zhan Wang[2,3], Dongning Zheng[2,3], Shuoming An [1✉], Yarui Zheng[1] & Shengyu Zhang[1]

Qubit initialization is a critical task in quantum computation and communication. Extensive efforts have been made to achieve this with high speed, efficiency and scalability. However, previous approaches have either been measurement-based and required fast feedback, suffered from crosstalk or required sophisticated calibration. Here, we report a fast and high-fidelity reset scheme, avoiding the issues above without any additional chip architecture. By modulating the flux through a transmon qubit, we realize a swap between the qubit and its readout resonator that suppresses the excited state population to 0.08% ± 0.08% within 34 ns (284 ns if photon depletion of the resonator is required). Furthermore, our approach (i) can achieve effective second excited state depletion, (ii) has negligible effects on neighboring qubits, and (iii) offers a way to entangle the qubit with an itinerant single photon, useful in quantum communication applications.

[1] Tencent Quantum Laboratory, Tencent, Shenzhen, Guangdong 518057, China. [2] Beijing National Laboratory for Condensed Matter Physics, Institute of Physics, Chinese Academy of Sciences, Beijing 100190, China. [3] School of Physical Sciences, University of Chinese Academy of Sciences, Beijing 100049, China. [4] These authors contributed equally. Yu Zhou and Zhenxing Zhang ✉email: shuomingan@tencent.com

Qubit initialization is fundamental and crucial for many quantum algorithms and quantum information processing tasks. The ability to quickly reset qubits to the zero state is one of DiVincenzo's essential criteria for building a quantum computer[1] and is critical for quantum error correction[2–4], where the reset of syndrome qubits needs to be accomplished with high fidelity in the time scale of a single qubit pulse. Furthermore, significant reduction of state preparation and measurement (SPAM) errors can be achieved by evacuating residual excited state populations with high fidelity[5,6]. The simplest way to reset qubits is to passively wait for them to de-excite, but as qubit relaxation times increase beyond 100 μs[7,8], this method becomes impractically slow. Alternatively, active reset implementations can shorten the wait time between cycles and significantly improve computational efficiency[9,10].

Various reset protocols for superconducting qubits have been proposed which fall into two main types: measurement- and non-measurement-based protocols. In measurement-based schemes, a qubit is measured and either heralded in the ground state[11], or else is found to be in the excited state and reset via a conditional π-pulse[6,12–15]. These protocols depend heavily on measurement fidelity and suffer from measurement-induced state mixing[5,16]. In addition, the hardware implementation of necessary short-latency feedback loops is also a challenge. In non-measurement based protocols, qubits are coupled to a lossy environment, usually a resonator. While numerous approaches to this have been proposed, they each suffer from their own drawbacks. For instance, in one such approach, flux control[17,18] is used to rapidly tune the qubit frequency to that of the resonator. However, this process significantly affects neighboring qubits via crosstalk[19,20]. Another approach is based on a microwave-induced interaction between the qubit and a low-quality factor resonator[9,21]. However, the involvement of the second excited state $|f\rangle$ makes these schemes complicated and necessitates sophisticated calibration. Furthermore, intense microwave driving is required to activate the required cavity-assisted Raman processes[21–23], affecting adjacent qubits as well. In[21], an additional resonator is required to achieve the best performance. In contrast to the above methods, the driven reset scheme proposed in[10] is free from flux control and complicated pulses. On the other hand, this protocol requires that the resonator dissipation rate $\kappa_r$ be smaller than the dispersive shift $\chi$, imposing a trade-off between readout speed and fidelity.

In this work, we demonstrate a rapid and unconditional parametric reset scheme for tunable superconducting qubits. By parametric modulation of the qubit frequency, a controllable interaction is generated between the qubit and a lossy readout resonator. This interaction unconditionally transfers the qubit excitation to the resonator and thus resets the qubit on demand. Using this method, we can suppress the residual excited population to $0.08\% \pm 0.08\%$ within 34 ns. We also demonstrate effective $|f\rangle$ state depletion in the case when leakage to higher states is non-negligible. Our protocol only involves AC modulation of at most two frequencies and does not need sophisticated calibration. Moreover, it has a negligible effect on subsequent gates and other qubits. It is compatible with circuit quantum electrodynamics systems[24–26] and can be applied to all frequency-tunable superconducting qubits, requiring no additional hardware or modifications to chip components. The method also imposes no restriction on operation flux position or specific system parameters such as resonator dissipation rate $\kappa_r$ or dispersive shift.

## Results

**Theory**. Our qubit reset protocol is based on a parametric activated interaction between a tunable qubit and a rapidly decaying resonator. Such a parametric modulation induces an effective tunable coupling between the qubit and other quantum systems such as another qubit or resonator[27–29] and has been used to implement multi-qubit quantum gates[30–35], state transfer[36,37], switches for quantum circuits[38] and parity measurements[39]. In our reset protocol, the parametric modulation induces Rabi oscillations between $|e, 0\rangle$ and $|g, 1\rangle$, where $|s, l\rangle$ denotes the tensor product of the qubit state $|s\rangle$ (the cases $|s\rangle = |g\rangle$ and $|s\rangle = |e\rangle$ correspond to the ground and excited states, respectively) and the resonator Fock state $|l\rangle$. When the qubit is excited, as illustrated in Fig. 1a, the population can be transferred from the qubit ($|e, 0\rangle$) to the resonator ($|g, 1\rangle$), which then rapidly decays to the target state $|g, 0\rangle$ at decay rate $\kappa_r$, which is mainly due to the large photon emission rate of the readout resonator.

We consider a qubit-resonator coupled system described by the Jaynes-Cummings model. In the dispersive regime, there is no population exchange due to the large detuning between the qubit and the resonator. The external flux $\Phi$ is modulated as $\Phi(t) = \overline{\Phi} + \Phi_m \cos(\omega_m t + \theta_m)$, where $\overline{\Phi}$ is the parking flux and $\Phi_m, \omega_m, \theta_m$ is the flux modulation amplitude, frequency and phase, respectively. Due to the nonlinear dependence of the qubit frequency on the flux bias, the qubit frequency $\omega_q(t)$ is, in general, described by a Fourier series with non-trivial higher-order terms, i.e. $\omega_q(t) = \overline{\omega_q} + \sum_{k=1} A_m^{(k)} \cos[k(\omega_m t + \theta_m)]$ where $A_m^{(k)}$ are the Fourier coefficients and $\overline{\omega_q}$ is the average frequency in the presence of the modulation[31]. In the case of small modulation, we take the leading term of the qubit frequency as an approximation, i.e. $\omega_q(t) \approx \overline{\omega_q} + A_m^{(\alpha)} \cos[\alpha(\omega_m t + \theta_m)]$, where $\alpha = 1$ for the qubit parked away from the sweet spot, and $\alpha = 2$ for the qubit parked in the sweet spot (in the latter case the odd Fourier coefficients $A_m^{(2k+1)}$ vanish[31]). The oscillation of the qubit frequency induces a series of sidebands $\overline{\omega_q} + n\omega_m$, where $n$ is an integer. When the frequency of one sideband satisfies the constraints $n\omega_m = -\overline{\Delta} = \omega_r - \overline{\omega_q}$, the transition between the states $|e, l\rangle$ and $|g, l + 1\rangle$ is activated. The effective coupling strength can be derived as $g_n = \overline{g_{qr}} J_n(\frac{A_m^{(\alpha)}}{\omega_m}) e^{i\beta_n}$, where $\overline{g_{qr}}$ is the averaged coupling strength between the qubit and the resonator during the modulation, $J_n(x)$ are Bessel functions of the first kind, and $\beta_n = n\theta_m - \frac{A_m^{(\alpha)}}{\alpha\omega_m}\sin(\alpha\theta_m)$ is the interaction phase[31].

We consider the single excitation subspace spanned by $\{|e, 0\rangle, |g, 1\rangle\}$, within which the dynamics of the reset protocol can be modeled by the non-Hermitian Hamiltonian

$$H_{\text{eff}} = \begin{bmatrix} 0 & |g_n|e^{i\beta_n} \\ |g_n|e^{-i\beta_n} & -i\kappa_r/2 \end{bmatrix}, \quad (1)$$

where $|g_n|$ is the absolute value of $g_n$, and the non-Hermitian term $-i\kappa_r/2$ accounts for the decay of the photon in the resonator. The population evolution can be expressed as $P_{s|s_0}(t) = |\langle s|e^{-iH_{\text{eff}}t}|s_0\rangle|^2$, where the system is initially prepared in the state $|s_0\rangle$, and $|s\rangle$ is one of the states $\{|e, 0\rangle, |g, 1\rangle\}$.

The real parts of the eigenvalues $\{\lambda_k\}$ of $H_{\text{eff}}$ determine the oscillation rate of $P_{s|s_0}(t)$, while the imaginary parts of $\{\lambda_k\}$ determine the exponential decay rates. We define the reset rate $\Gamma = 2\min_k(|\text{Im}[\lambda_k]|)$ as it is the smallest value of the decay rates and determines the overall protocol reset speed. Three different regimes are possible – corresponding to overdamped, critically damped and underdamped oscillations of $P_{s|s_0}(t)$, respectively – and our qubit reset works in all three regimes. For small modulation amplitudes, i.e. $|g_n| < \kappa_r/4$, the reset is in the overdamped regime where the excited state population decays without oscillating. In this

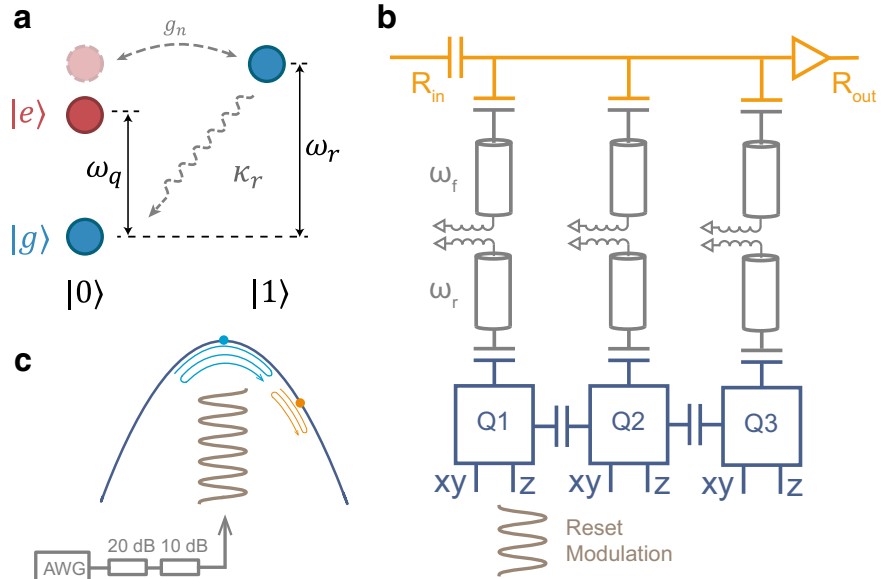

**Fig. 1 Schematic of the reset process and diagram of the device. a** Jaynes-Cummings ladder diagram of the qubit-resonator coupled system. $|g\rangle$, $|e\rangle$ and $|0\rangle$, $|1\rangle$ denote the qubit and resonator lowest two states respectively. The dashed light red circle represents one of the sideband modes induced by the parametric modulation. The dashed arrow labeled $\kappa_r$ illustrates the decay process of the resonator. **b** Simplified circuit diagram of the chip. Each transmon qubit is coupled to a readout resonator and an individual Purcell filter. The readout signal is amplified by an impedance-matched Josephson parametric amplifier (IMPA). **c** Typical transmon resonance frequency $\omega_q$ with respect to the flux bias. The reset pulse is generated by AWG. After 30 dB attenuation, it is added to the flux control line. The qubit's frequency modulation (brown) activates effective coupling between the qubit and its readout resonator and facilitates the reset process. Two cases are depicted: (1) operation point at or near the sweet spot (cyan, main text); (2) operation point away from the sweet spot (orange, Supplementary Note 5).

regime, the reset rate $\Gamma$ increases with the modulation amplitude. At the critically damped point $|g_n| = \kappa_r/4$, the population shows a maximum reset rate $\kappa_r/2$ with no oscillation. When the modulation amplitude satisfies $|g_n| > \kappa_r/4$ the reset becomes underdamped, and the population oscillates at rate $\sqrt{4|g_n|^2 - \kappa_r^2/4}$, and the reset rate remains at $\kappa_r/2$.

**Experimental realization.** Our experimental setup is depicted in Fig. 1b and consists of three transmon qubits[40,41]. Each transmon is capacitively coupled to a resonator with frequency $\omega_r/2\pi$ from 6.44 GHz to 6.68 GHz and coupling strength $g_{qr}/2\pi$ around 80 MHz. Individual Purcell filters[42,43], implemented by $\lambda/4$ resonators, are inductively coupled to each readout resonator, and XY control and flux control (Z) lines are coupled to each qubit. Fig. 1c displays the frequency $\omega_q$ of a transmon qubit with respect to the flux. The reset pulse is generated from an arbitrary waveform generator (AWG). After 30 dB attenuation, it is fed into the Z line, which results in frequency modulation of the qubit.

We first demonstrate parametric reset on isolated Q1 (Q2, Q3 are tuned to their minimal frequencies through a fixed DC bias). Fig. 2a shows the detailed sequence: A $\pi$-pulse is applied through the XY driveline to prepare the qubit in state $|e\rangle$. A sinusoidal parametric reset pulse $A\sin(\omega_m t)$ of duration 1000 ns is applied through the Z line, with amplitude $A$ and frequency $\omega_m$. Finally, Q1 is measured by the traditional dispersive readout. Fig. 2b shows the measured $|e\rangle$ population after this reset process, as a function of modulation amplitude $A$ (displayed in units of the magnetic flux quantum $\Phi_0$).

Several strips—labeled $n = 1, 2, 3$, and corresponding to $n$-th order modulations—are visible, where the population of $|e\rangle$ drops dramatically compared to other regions. In these regions, one of the qubit's modulation sidebands is close to the resonator frequency, and the population transfers from the qubit to the resonator. When the operation point ($\omega_q/2\pi = 5.784$ GHz, $\omega_r/2\pi = 6.441$ GHz, $\Delta/2\pi = -0.659$ GHz) of the transmon qubit is close to the sweet spot, the qubit frequency will undergo two oscillations for every cycle of the parametric drive. As previously described in the theory section, the actual qubit modulation frequency is thus $2\omega_m$, twice that of the flux modulation frequency $\omega_m$. There are several thin and unmarked strip-shaped regions in the figure due to the imperfect match between the operation point ($0.004\ \Phi_0$) and the sweet spot. It is worth noting that sweet spot operation is not a requirement for our parametric reset protocol, and non-sweet spot operation is also suitable (see Supplementary Note 5). Three points A, B, C in the $n = 1$ region were selected and, for each point, the qubit was first prepared in the $|e\rangle$ state and the population $p_e$ of $|e\rangle$ was then measured as a function of the duration of the parametric reset pulse $\tau$ by direct readout measurements (see Fig. 2c). Corresponding to small modulation amplitude, point A (blue) lies in the overdamped regime where the $|e\rangle$ state population decays slowly and without any oscillation. At point C (red), the modulation amplitude is large, and the $|e\rangle$ state population oscillates heavily, corresponding to the underdamped regime. Solid lines are fit to the theoretical model for $P_{s|s_0}(t)$ (Supplementary Note 4). From the fitting, we extract $\kappa_r^{-1}$ of 46 ns, which agrees with the direct measurement of the photon decay ($\kappa_r^{-1} \approx 50$ ns) of the resonator via AC Stark shift[44]. The insert displays $\Gamma/\kappa_r$ vs. $|g_n|/\kappa_r$ predicted by theory, where $\Gamma$ is the reset rate for qubit state $|e\rangle$. $\Gamma/\kappa_r$ increases in the overdamped regime and saturates at $\kappa_r/2$ in the underdamped regime. Results for point B, chosen to be close to the critically damped point, are displayed in green in Fig. 2c and show a population decay much faster than in the overdamped regime, with no oscillations observed. A master equation simulation was performed of the whole process using all experimental parameters, and the results are shown in Fig. 2d. Due to the limited sampling rate of the AWG, the amplitude of the modulation signal is heavily

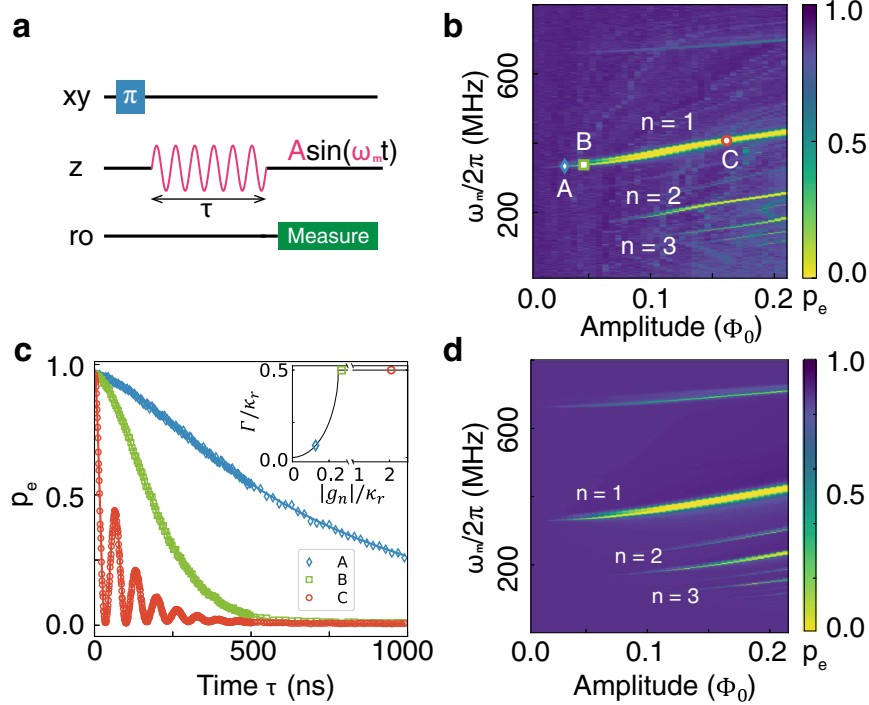

**Fig. 2 Realization of the parametric reset on Q1. a** Experimental sequence of the parametric reset. A $\pi$-pulse is applied to the qubit and followed by a sinusoidal parametric reset pulse. The amplitude $A$ and angular frequency $\omega_m$ are two adjustable parameters. **b** Two dimensional scan of the $|e\rangle$ population, $p_e$ when $\omega_q/2\pi = 5.784$ GHz, $\omega_r/2\pi = 6.441$ GHz, $\Delta/2\pi = -0.659$ GHz. The x-axis is the parametric amplitude in the magnetic flux quantum $\Phi_0$ and the y-axis is the modulation frequency. First, second, and third-order modulations are labeled $n = 1, 2, 3$, respectively. **c** Time evolution of the excited state population, after parametric modulation corresponding to the three points A, B, C in the $n = 1$ region in **b**. Dots are the raw data acquired by direct readout measurements, and solid lines are fits to the theoretical model. The insert shows theoretical values of $\Gamma/\kappa_r$ vs. $|g_n|/\kappa$, where $\Gamma$ is the reset rate of the qubit during the reset process. **d** Master equation simulation of the whole process with the same parameters as the experimental (**b**).

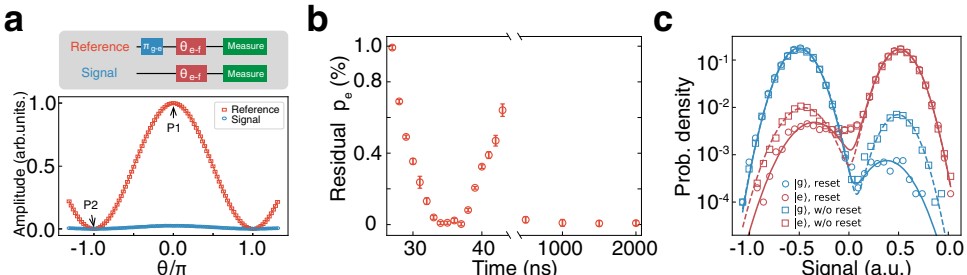

**Fig. 3 Residual $|e\rangle$ state population measurement and improvement of the readout fidelity. a** Residual $|e\rangle$ population measurement with Reference (red) and Signal (blue) Rabi oscillations with solid sinusoidal fitting curves. From the fitting, two Rabi amplitudes($A_{sig}$ and $A_{ref}$) can be extracted. The Reference and Signal pulse sequences are shown in the upper left corner. P1 and P2 are two measured points in **b**. **b** $|e\rangle$ population measurement with different parametric reset pulse duration $\tau$. Each data point is acquired by a "two-point method" described in the main text. At 34 ns, the population of the $|e\rangle$ decays to the first minimum value 0.08 ± 0.08% and remains below 0.1% after 1000 ns. The error bars are statistical ( ± 1 s.d.) with 50 repetitions. **c** Readout fidelity enhancement with the 34 ns parametric reset pulse. The circles (squares) display the IQ analysis with (without) the 34 ns parametric pulse. Both $|g\rangle$ (blue) and $|e\rangle$ (red) are prepared and measured. When preparing the $|g\rangle$ state, the residual $|e\rangle$ state population due to the thermal excitation decreases more than one order of magnitude after applying the parametric reset pulse. The readout fidelity consequently improves from 96.13% to 99.43% for $|g\rangle$ and 92.69% to 96.05% for $|e\rangle$ respectively.

attenuated by the analog reconstruction filter at a high frequency. We reproduce this lowpass filter effect of the AWG in our simulation and find the main features agree with the experiment very well. In situations where high-frequency modulation is a must (e.g., in the case of large detuning between the qubit and resonator and where the first-order region is preferred), the AWG can be replaced with a microwave source to overcome this limitation.

To measure the residual $|e\rangle$ state population, we perform a Rabi population measurement (RPM)[10,45] involving both the $|e\rangle$ and $|f\rangle$ (second excited) states in the sequence shown in Fig. 3a. To acquire the Reference data (red squares), we first apply a $\pi$-pulse to flip the population between the $|g\rangle$ and $|e\rangle$ states, then perform a rotation around $X$ between $|e\rangle$ and $|f\rangle$ of angle $\theta$ ($\theta$-pulse). Varying the angle $\theta$ results in Rabi oscillations of the $|e\rangle$ state population, as the Reference (red) shows. The solid line is a sinusoidal fitting. For the Signal data (blue circles), no $\pi$-pulse between $|g\rangle$ and $|e\rangle$ is applied. However, there is still a visible and relatively small Rabi oscillation due to the residual thermal $|e\rangle$

population. The final portion of residual $|e\rangle$ population $p_e$ is calculated by $A_{sig}/(A_{sig} + A_{ref})$, where $A_{ref}$ and $A_{sig}$ are the fitted amplitudes of the red and blue oscillations, respectively. Without any parametric reset, the measured $|e\rangle$ residual excited state population is around $2.38 \pm 0.06\%$ which corresponds to a 75 mK effective temperature.

We attribute this relatively high effective temperature to stray infrared radiation and insufficient thermalization of the sample box to the cold finger, which can be improved by careful shielding and better thermal contact[45,46]. The time evolution of point C is shown in Fig. 2c. When $\tau = 34$ ns, the $|e\rangle$ state population decays to its first minimum and reaches a steady low level after 1000 ns. Both the first minimum and steady-state points are of practical significance. The former is useful when reset speed is the priority, and the reset protocol does not need to be reused immediately; the latter is insensitive to parametric modulation time and requires fewer calibrations. To reduce measurement error and accurately deduce the $|e\rangle$ residual population after the parametric reset pulse, a "two-point method"[45] was used to increase the data acquisition efficiency. Instead of measuring the whole trace as in Fig. 3a, only the maximum and minimum points of the oscillation — marked as P1, P2—were measured ($2 \times 10^5$ times each) to determine each value of $p_e$. Fig. 3b shows the residual population of excited state $|e\rangle$ after variable parametric reset duration $\tau$ deduced via this two-point method. Each point corresponds to 50 measurements of $p_e$ with one standard deviation error bar. The minimum residual population reaches $0.08 \pm 0.08\%$ at 34 ns and remains below 0.1% after 1000 ns, outperforming all existing reset schemes (Supplementary Note 3). Based on the rate model from Supplementary Note 6, we estimate the residual excitation population to be 0.02% in thermal equilibrium.

The parametric reset process decreases state preparation error, yielding better readout fidelity, as it effectively reduces the thermal population as illustrated by Fig. 3c, where circles (squares) represents the measurement with (without) the 34 ns parametric reset pulse. With the parametric reset, the thermal excitation $p_e$ is reduced by more than an order of magnitude when preparing the $|g\rangle$ state. Performing state discrimination analysis yields a significant improvement of the readout fidelity with $|g\rangle$ from 96.13% to 99.43% and $|e\rangle$ from 92.69% to 96.05%, respectively.

**Reset of the $|f\rangle$ state by two-tone parametric drive.** Leakage to the second excited state $|f\rangle$ can be an important source of error during two-qubit gates[47] and measurements[48]. In this section, we extend the single-tone parametric modulation scheme to one that uses two-tones in order to achieve effective $|f\rangle$ state reset. In this case, the reset pulse has the form $A_1 \sin(\omega_1 t) + A_2 \sin(\omega_2 t)$, where $A_1, A_2$ and $\omega_1, \omega_2$ are the amplitudes and frequencies of the two tones used, respectively, and the corresponding Fourier expansion of the qubit frequency has four main frequency components: $2\omega_1, 2\omega_2, \omega_1 \pm \omega_2$. As shown in Fig. 4a, depletion of the $|f\rangle$ state comprises two processes. First, one frequency component of the parametric modulation causes $|f, 0\rangle$ to interact with $|e, 1\rangle$, with the latter decaying to $|e, 0\rangle$ at the resonator dissipation rate $\kappa_r$. Similarly, $|e, 0\rangle$ decays to $|g, 0\rangle$ via the second frequency component of the modulation. Fig. 4b is a scanned map of the $|e\rangle$ state population after a 1000 ns two-tone parametric reset pulse, with the qubit initially prepared in the $|f\rangle$ state. We consider the case where $A_1 = A_2$, and scan $\omega_1/2\pi$ and $\omega_2/2\pi$ from 230 MHz to 730 MHz. Two spider-like regions can be seen in the figure, one consisting of multiple blue strips and the other consisting of multiple yellow strips. These two regions correspond respectively to the first and second decay processes mentioned above. Six of

these colored strips are annotated in the figure, where strips 1,2 (4,5) correspond to the regions where $2\omega_{1(2)} = -\overline{\Delta}$ ($2\omega_{1(2)} = -\overline{\Delta} - \eta$), and strip 3 (6) corresponds to the region where $\omega_1 + \omega_2 = -\overline{\Delta}$ ($\omega_1 + \omega_2 = -\overline{\Delta} - \eta$). Here, $\eta = -254$ MHz is the anharmonicity of Q1. We perform a master equation simulation of this two-tone parametric reset process and find that the results (Fig. 4c) closely agrees with the experiment. See Supplementary Note 8 for more scan maps and theoretical explanation. In the rhombus area marked R in Fig. 4b, where strips 1 and 6 intersect, the two decay processes coexist, and the region is thus suitable for fast depletion of the $|f\rangle$ state. In Fig. 4d-e we consider the case where $A_2 = 1.8A_1$ and measure the time evolution of the $|g\rangle, |e\rangle, |f\rangle$ states in the R region. Circular data points in these figures are experimental data, and the solid lines fit to a multi-level decay model[49]. The in-phase(I) and quadrature(Q) components of each state from the readout are shown inset in Fig. 4e. The qubit was prepared in the $|e\rangle$ (Fig. 4d) and $|f\rangle$ (Fig. 4e) states, respectively, and the population of all excited states $1 - P_g$ is shown in Fig. 4f. The excited population for both initial states shows nearly exponential decay and reaches the readout floor within 600 ns (initial state: $|e\rangle$) and 1000 ns (initial state: $|f\rangle$). From the multi-level decay model[49], we estimate the decay rates to be 1/100 ns for state $|e\rangle$ and 1/117 ns for $|f\rangle$ during the reset process. The measured reset fidelity is 99.23%, limited by the readout fidelity.

**Scalability of the protocol.** To study the protocol's scalability, parametric resets were simultaneously performed on two qubits Q1 and Q2. In this case, all qubits were tuned to operation points near their sweet spots. The sequence is shown in the top part of Fig. 5a. Both qubits are prepared in the $|e\rangle$ state by a $\pi$-pulse, and parametric modulation is applied separately to each qubit through their associated flux lines. The time evolution of the excited state population $p_e$ of the qubits is measured with varied reset duration $\tau$. As seen in Fig. 5a, $p_e$ decays quickly and remains at a low level from 2 $\mu$s onwards, demonstrating the feasibility of our parametric reset protocol in a multi-qubit system. One disadvantage of previous reset protocols involving flux pulses[17,18] is that the Z pulse for the reset can significantly affect all subsequent gates[20,50–52] or cause a frequency shift of neighboring qubits[53]. The parametric modulation we propose consists of one or two frequency components only and thus has negligible effects on neighboring qubits. To prove this, Clifford-based randomized benchmarking (RB) was performed on Q2(Q3) when Q1 was reset with a single-tone parametric pulse (the results of RB with two-tone reset are given in Supplementary Note 10). From the RB data (Fig. 5b), we find that the reset process decreases the average gate fidelity by only 0.08% for nearest neighbor qubit Q2, and has almost no effect on the next-nearest neighbor qubit Q3, with only 0.03% fidelity difference. To further probe the effect of the reset process on the neighboring qubits, we have also performed a series of Ramsey measurements, with results given in Supplementary Note 7. Together, these experiments demonstrate that our parametric reset protocol has a negligible effect on adjacent qubits in terms of coherence, frequency and gate fidelity.

**Discussion**

In our single-tone parametric scheme, the first minimum point has practical significance when there is no immediate gate on the same qubit or when gates are immediately applied to other qubits, such as in state transfer[54] or quantum simulations[55]. In scenarios where photon depletion of the resonator after qubit reset is a must, a $5/\kappa$ (250 ns) duration passive wait is sufficient to deplete the mean photon number below 0.01, corresponding to a

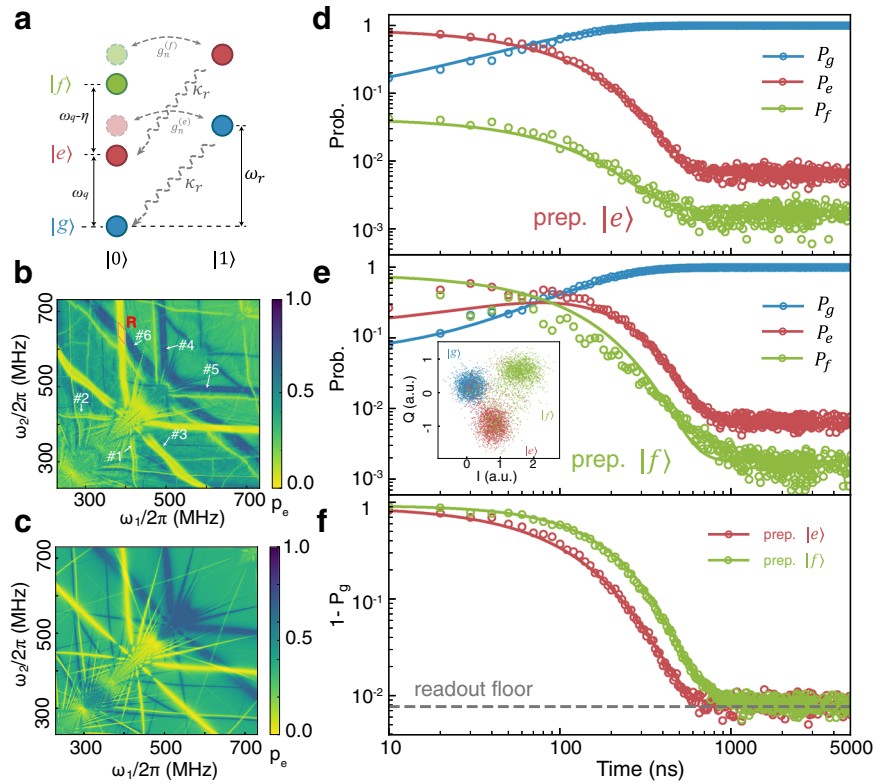

**Fig. 4 Two-tone parametric modulation. a** Jaynes-Cummings ladder diagram of the two-tone parametric modulation process. $|g\rangle$, $|e\rangle$, $|f\rangle$ and $|0\rangle$, $|1\rangle$ denote the lowest energy states of the qubit and resonator, respectively. The dashed pale red and green circles represent one of the sideband modes induced by the parametric modulation. The dashed arrow labeled $\kappa_r$ illustrates the decay process of the resonator. **b** Scan map of the $|e\rangle$ population after 1000 ns two-tone parametric modulation. Before modulation, the qubit was prepared in the $|f\rangle$ state. Blue and yellow strips correspond to the $|f,0\rangle \rightarrow |e,1\rangle \rightarrow |e,0\rangle$ and $|e,0\rangle \rightarrow |g,1\rangle \rightarrow |g,0\rangle$ decay processes, respectively. **c** Master equation simulation with experimental parameters from **b**. Time evolution of each qubit state, with the qubit initially prepared as $|e\rangle$ (**d**) or $|f\rangle$ (**e**), and then reset with two-tone modulation. Inset figure in **e** is the IQ plot of each state. **f** Time evolution of all remaining excited states ($1 - P_g$). The gray dashed line is the readout floor.

negligible stark shift of 51.5 kHz. Taking re-thermalization (Supplementary Note 6) into consideration, the net fidelity at the end of this process is 99.86%. An alternative way to achieve photon depletion of the resonator is to reset the resonator actively[56,57]. If speed is not a priority, a one-tone reset with the modulation always on is an option, requiring 1000 ns and achieving 99.9% fidelity. When $|f\rangle$ state leakage is not negligible, two-tone modulation is preferable and it takes 1000 (600) ns for both the $|e\rangle$ and $|f\rangle$ state ($|e\rangle$ only) to reach the 0.77% readout floor. In conclusion, the parametric reset schemes we proposed have a high degree of flexibility that allows them to be used in a variety of different scenarios. We summarize the performance and use case scenarios of our protocols in Supplementary Table 2.

We have demonstrated a parametric reset protocol realized in transmon qubits which can be completed in 34 ns (284ns if one waits an additional five times the resonator $T_1^r = 1/\kappa_r$ time for the resonator to deplete). The speed and fidelity 99.92% (99.86% if $5/\kappa_r$ is included) of our approach outperforms all existing reset schemes (Supplementary Note 3) and, furthermore, has the added advantages of flexibility and scalability. In theory, the reset time can be further decreased to less than 10 ns by increasing the modulation amplitude of the reset pulse or by increasing the coupling strength $g_{qr}$ in the qubit design. Moreover, as the RB and the Ramsey experiments show, our parametric modulation induces negligible effects on neighboring qubits in terms of gate fidelity, frequency and coherence. By extending the method to using two-tone modulation, we are also able to achieve effective $|f\rangle$ state depletion. Our methods give a practical and universal way to reset tunable superconducting qubits and offer a pathway

to achieving high-fidelity reset in large-scale qubit systems. Beyond qubit reset, parametric modulation-induced interaction can also be used in thermodynamic reservoir engineering[37,58] and quantum many-body simulations[55]. Furthermore, this work provides an efficient way to entangle the qubit state with an itinerant single photon, particularly useful in quantum communication and quantum network application[54,59,60].

## Methods

**Hamiltonian with parametric modulation**. We consider a qubit-resonator coupled system, which can be described by the Jaynes-Cummings model ($\hbar = 1$ hereafter)

$$H_{\text{sys}} = \omega_q |e\rangle\langle e| + \omega_r a^\dagger a + g_{qr}(a^\dagger \sigma_- + a\sigma_+), \quad (2)$$

where $\omega_q$ ($\omega_r$) is the qubit (resonator) frequency, $g_{qr}$ is the coupling strength between the qubit and the resonator and $\sigma_+$ ($\sigma_-$) is the creation (annihilation) operator of the qubit. In the interaction picture, when the qubit frequency is modulated as $\omega_q(t) \approx \overline{\omega_q} + A_m^{(\alpha)}\cos[\alpha(\omega_m t + \theta_m)]$ ($\alpha$ an integer), to leading-term approximation the system Hamiltonian can be expressed as

$$H_{\text{int}} = \sum_{n=-\infty}^{\infty} g_n e^{i(n\alpha\omega_m + \overline{\Delta})t} a\sigma_+ + h.c. \quad (3)$$

where $g_n = \overline{g_{qr}} J_n\left(\frac{A_m^{(\alpha)}}{\alpha\omega_m}\right) e^{i\beta_n}$ are the effective coupling strengths in the leading term, $\overline{g_{qr}}$ is the averaged coupling strength during the modulation, $\beta_n = n\alpha\theta_m - \frac{A_m^{(\alpha)}}{\alpha\omega_m}\sin(\alpha\theta_m)$ is the interaction phase, and $J_n(x)$ are Bessel functions of the first kind, and $\overline{\Delta} = \overline{\omega_q} - \omega_r$ is the effective detuning between the qubit and the resonator during the sinusoidal modulation[31]. Note that for transmon qubits, the modulation of the qubit frequency also induces a modulation of the coupling strength, the full expression of which can be found in reference[31].

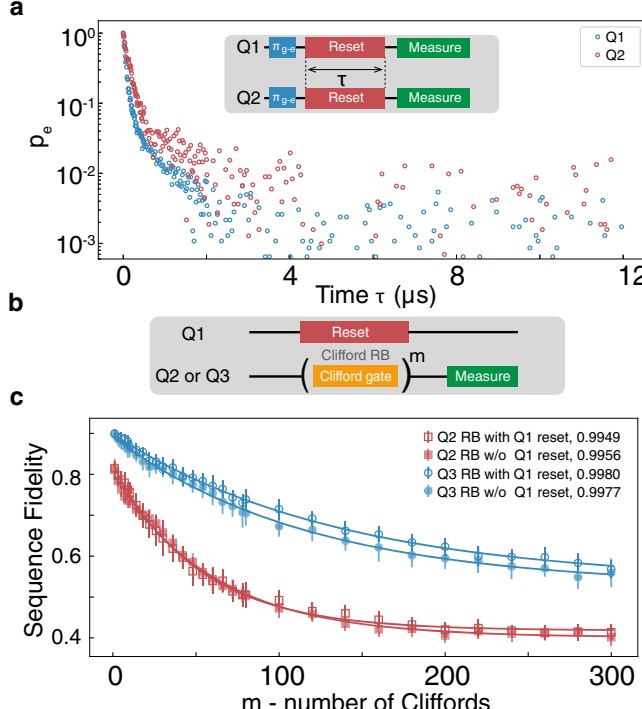

**Fig. 5 Simultaneous parametric reset of Q1, Q2, and effects of reset pulse on adjacent qubits. a** Time evolution of Q1 (blue) and Q2 (red) $|e\rangle$ state population when performing a parametric reset on both qubits simultaneously. The population remains at a steady low level from 2 $\mu$s onwards. The pulse sequence for simultaneous parametric reset is inset. **b** Pulse sequence for Clifford-based randomized benchmarking (RB) on Q2(Q3). During the RB, either a parametric reset (with the same pulse duration as the $m$ Clifford gates) was applied to Q1 or, for comparison, no pulse was applied to Q1. **c** RB results for the scenario described in **b**. The error bars are statistical ($\pm 1$ s.d.) with 30 repetitions. The average single gate fidelity standard deviation is 0.08%(0.03%) for Q2(Q3).

When the modulation frequency $\omega_m$ satisfies the constraint $n\alpha\omega_m + \overline{\Delta} = 0$ (for integer $n$) the Hamiltonian approximates to $H_{\text{int}} = g_n a\sigma_+ + g_n^* a^\dagger \sigma_-$ by ignoring rapidly oscillating terms, and Rabi oscillations occur between states $|e, l\rangle$ and $|g, l+1\rangle$.

**Qubit reset rate**. When the qubit is prepared in the excited state $|e\rangle$, the time dependent population can be solved by the effective Hamiltonian $H_{\text{eff}}$ of equation (1). The reset rate $\Gamma$ can be derived as:

$$\Gamma = 2\min_k(|\text{Im}[\lambda_k]|) = \begin{cases} \frac{1}{2}\left(\kappa_r - \sqrt{\kappa_r^2 - 16|g_n|^2}\right) & |g_n| < \kappa_r/4 \\ \kappa_r/2 & |g_n| \geq \kappa_r/4 \end{cases} \quad (4)$$

For $|g_n| \geq \kappa_r/4$, the reset rate remains at $\kappa_r/2$ and is independent of the effective coupling $|g_n|$. For $|g_n| < \kappa_r/4$, the reset rate $\Gamma$ increases with the effective coupling. In both cases, the reset rate is larger than the free decay rate of the qubit. The population of the qubit during the parametric reset process can be modeled by

$$P_{s|s_0}(t) = |\langle s| \exp(-iH_{\text{eff}}t)|s_0\rangle|^2, \quad (5)$$

where the system is initially prepared in the state $|s_0\rangle$, and $|s\rangle \in \{|e, 0\rangle, |g, 1\rangle\}$. When $|s_0\rangle = |e, 0\rangle$, the population of the excited state $p_e$ can be shown to be

$$p_e = P_{e|e}(t) = \begin{cases} e^{-\frac{\kappa_r t}{2}}\left(\frac{\kappa_r t}{4} + 1\right)^2 & |g_n| = \kappa_r/4 \\ e^{-\frac{\kappa_r t}{2}}\left[\cos(Mt) + \frac{\kappa_r}{4M}\sin(Mt)\right]^2 & |g_n| > \kappa_r/4, \; M = \frac{\sqrt{16|g_n|^2 - \kappa_r^2}}{4} \\ e^{-\frac{\kappa_r t}{2}}\left[\cosh(Mt) + \frac{\kappa_r}{4M}\sinh(Mt)\right]^2 & |g_n| < \kappa_r/4, \; M = \frac{\sqrt{\kappa_r^2 - 16|g_n|^2}}{4} \end{cases} \quad (6)$$

## Data availability

Source data to generate figures and tables are available from the corresponding authors.

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

## Acknowledgements

We acknowledge the support from the Key-Area Research and Development Program of Guangdong Province (2020B0303030002 and 2020B0303030001), the State Key Development Program for Basic Research of China (Grant No. 2017YFA0304300) and the Strategic Priority Research Program of Chinese Academy of Sciences (Grant No. XDB28000000).

## Author contributions

S.M.A. and X.G. conceived the experiment and developed the theory. Y.Z., Z.X.Z. and S.N.H. built the set-up and carried out the measurement. Z.L.Y. and Z.X.Z. performed numerical simulations. Y.Z., Z.X.Z., S.M.A. and Y.R.Z. analyzed the results. Y.Z. and Z.X.Z. wrote the manuscript. J.A. and S.M.A. edited the manuscript. Z.X.Z. and X.X. developed the software platform for the experiment. H.K.L., X.H.S., Z.W. and D.N.Z. prepared the sample. F.M.L., G.L.X., Q.N.Y., M.Y.Z. and H.L.Z. developed room-temperature electronics. S.Y.Z. supervised the project. All authors contributed to the discussion of the results and the development of the manuscript.

## Competing interests

The authors declare no competing interests.
