## [Peer Review File · Nature Communications]

Reviewers' Comments:

Reviewer #1:

Remarks to the Author:

The authors demonstrate an unconditional qubit reset method, using a parametrically modulated flux pulse and a dissipative resonator. They use it on a 3-superconducting qubit device, demonstrating the reset of up to two qubits simultaneously, without significantly affecting the remaining one(s).

I have concerns about the analysis of the reset fidelity, which only assumes that the initial state is the first qubit excited state, and disregards the state of the resonator at the end of the reset pulse.

* The methods used to measure the ground (P_0) and first excited state population (P_1) [10 and 46 in the manuscript] both assume that higher level populations (P_2 or above) are negligible. From the estimated effective temperature of 75 mK, this is a reasonable assumption. However, it is widely reported that both two-qubit gates [Marques et al., arXiv:2102.13071] and readout [18] are notable sources of leakage to $|2\rangle$, which accumulates significantly throughout several error correction rounds. Therefore the claim that $|2\rangle$ is negligible cannot be substantiated beyond the steady (thermal) state.

* The protocol is only effective for initial state $|1\rangle$. The theory also addresses only the qubit subspace. Would it be possible to apply the same method to deplete $|2\rangle$, and subsequently deplete $|1\rangle$? If so, what time / fidelity would be expected?

* The highest speed of the reset pulse is measured by timing the reset pulse to the first minimum in the qubit $|e\rangle$ population. Although this is the ground state for the qubit, it is not the ground state for the readout resonator, as can be seen by the revival of qubit population over multiple exchanges with the cavity (Fig.2c) In order for any operation on the qubit to succeed with high fidelity or without dephasing induced by fluctuations in the photon number, the resonator must first be emptied. In the absence of active depletion of the resonator, one needs to wait $\sim 5\kappa$ for the reset operation to be complete, similarly to the case when the flux pulse is left on throughout the whole time. If the authors motivate this technique in the context of repeated error correction (introduction), resetting both qubit and resonator before the next operation would take at least 800 ns (Fig.3b), which is not faster than other methods (as seen on Fig. S2).

For the above reasons, I find the claim that this method "outperforms all existing reset schemes" (as stated in the conclusion and in Fig. S2) hardly justifiable. There is not enough information to quantify the ability to prevent or suppress leakage to higher excited states, and the reset time (for the intended application) does not appear to be any shorter than the cited methods.

Other minor comments. In the main text:

* In the introduction "the syndrome qubits need to be reset on demand". Technically, reset of the ancillas at every round of error correction round is not necessary (although definitely helpful in reducing leakage accumulation, see main comments above)

In the Supplementary Material:

* "The population of the population" (beginning of note 4)

* Fig. S2, some of the years are wrong: (Geerlings et al, 2013; Riste et al, 2012)

* I find the distinction between active control and feedback control confusing, as feedback is a type of active control. Perhaps measurement and non-measurement based, or unconditional would be more clear.

Reviewer #2:

Remarks to the Author:

The authors describe and experimentally demonstrate a method for resetting a transmon superconducting qubit by means of a parametrically induced coupling to its readout resonator. Reliable active reset protocols such as this one are very important in the operation of

superconducting quantum computing systems. The presented protocol is conceptually rather simple and I believe that many superconducting circuit labs have explored and tested similar approaches (swapping an excitation from the qubit to a resonator by parametric driving). However, to my knowledge, a clear exposition of such tests together with detailed quantitative analysis has so far been missing from the published literature.

The manuscript is well written, the explanation of the underlying principles clear and the experimental results well presented. I believe the scheme and its presentation would be highly relevant to researchers in the field of quantum computing with superconducting qubits and it warrants publication in Nature Communications with a few minor revisions.

1) The authors refer to the recent work by McEwen et al. [18] as requiring careful calibration of flux pulses to achieve tuning of the qubit on resonance with the readout resonator. To my understanding, this is not exactly correct since the cited work tunes the qubit through resonance with the resonator to exchange the excitation adiabatically, which does not require careful adjustment of the frequency.

Also, I am not sure if it is appropriate to represent the results of [18] in Fig. S2 by a square (active control at first minimum) rather than a circle (active control at 8τ). The 200-300ns shown in Fig. S2 by the data-point for [18] correspond to about 6 times the resonator relaxation time (45ns) and there are not really any minima to speak of since the protocol does not rely on resonant swapping. I think it would be fairer to show a circle for [18] (possibly showing a longer protocol time $8\tau = 360$ ns).

2) Since the authors claim quite prominently that their method outperforms all existing reset schemes, I think Fig. S2 which backs this claim (together with a short discussion) would deserve to be in the main text of the manuscript rather than in the supplementary material. I also believe it could be stated more explicitly that the first minimum points (indicated by squares in Fig. S2) represent the reset time only in settings where one does not need to reuse the protocol immediately. Such settings certainly exist and in these, the very fast excitation swap the authors demonstrate would have its use. However, in many other cases, one would still need to wait for the excitation in the resonator to decay before another reset can be performed with the same resonator. This sets an effective minimum time several times larger than $1/\kappa$. The authors correctly mention this in the discussion section but I think it is a rather important aspect that could be mentioned more prominently.

3) Could the authors explain why the decay rate of the excited state population in curve B in Fig. 2c seems to be much slower than might be expected from the stated κ_r ? If $1/\kappa_r$ is approximately 50 ns and therefore $1/\Gamma = 100$ ns, shouldn't we expect the population to decrease to approximately 0.37 in about 100 ns?

4) Have the authors looked at how the protocol acts on initial states other than the excited state? Specifically, if the qubit is in the ground state to begin with, is there any chance that the protocol actually excites it? I would not expect this to happen in a parametric scheme (which should preserve the total number of excitations) but in my opinion it would be important to verify this explicitly. If the authors have investigated this, a brief sentence stating the highest observed excitation probability (or if this was unmeasurable, mentioning it explicitly) may be enough to address this.

5) Have the authors analyzed the source of the background residual population? 2.4% (equivalent to 75mK) seems rather high and I wonder if this could be due to insufficient thermalization of some of the input lines. I would see this as a problem if this is caused by the rather low attenuation of the flux lines (Fig. S1 shows no attenuators in them), since I expect the rather strong parametric driving required for this protocol may not be compatible with higher attenuation. Can the authors comment on whether the effective temperature is limited by the flux line attenuation and whether that is in turn dictated by the required strength of the parametric drive?

6) I am curious about the statement that "...residual population reaches 0.08%±0.08% at 34ns and remains below 0.2% after 800ns". Does this mean the population rises from the 0.08%

achieved right after the reset pulse to 0.2%? If so, can the authors say what could be causing this? Is it just the spontaneous re-thermalization of the qubit with its effective 75 mK environment?

Reviewer #3:

Remarks to the Author:

The manuscript by Zhou et al. convincingly demonstrates the functionality described in the title, that is, a rapid and unconditional parametric reset protocol for frequency-tunable superconducting qubits. The protocol is based on parametric modulation of the frequency of one of two coupled modes with direct coupling and large detuning, a technique that has been extensively studied before, but not applied to qubit reset. Existing literature on the subject is well acknowledged and a critical comparison is drawn between present and previous realizations. Advantages of this reset protocol are its speed (the fastest to-date in the "timed mode"), absence of hardware overhead (the resonator used for the reset doubles as a readout resonator, same as in Magnard2018), and simplicity of operation. Possible disadvantages are that it only applies to frequency-tuneable qubits, and possibly that – at least for the parameters explored here – a Purcell filter is needed to achieve coherence times larger than a few microseconds, due to the relatively large coupling strength and relatively small detuning between qubit and resonator.

The analysis of the reset protocol is scientifically sound. However, I was not convinced by the investigation of the effects of the reset protocol on neighboring qubits (see first comment below). The theory explanation of sideband driving could also be improved. In summary, the proposed scheme is a valuable addition to the existing reset schemes in the community, but I do not regard it as a major advance. For this reason, I am unsure whether this manuscript belongs to Nature Communications as opposed to a more specialized journal. In any case, I recommend that the authors address the comments below.

1. Of your three qubits, Q2 and Q3 have very bad dephasing times ($< 1\mu\text{s}$), while Q1 has a better one ($\sim 10\mu\text{s}$). Can you comment on possible reasons for this difference? In addition, when looking at the effects of the reset protocol on neighboring qubits, it would make most sense to analyze the dephasing induced on your "best" qubit, Q1, while trying to reset Q2 (which you show you can do in Fig. 4b). Instead, you are only considering the qubits with very bad dephasing time and show that resetting the "good" qubit does not make it even worse. That does not seem like the best comparison you can make, and certainly does not justify your claim that the method "has negligible effect on neighboring qubits", also given that state-of-the-art coherence times (T_2) for superconducting qubits are of the order of $100\mu\text{s}$, not $1\mu\text{s}$.

2. L84, the statement "A sinusoidal modulation of the external flux..." is incorrect because of the nonlinear relationship between frequency and flux. In fact, in L128 you correct yourself with a statement that the non-specialist reader may not understand. Please follow the standard treatment of this process in which you start from the full expression and then take a first and second order expansion in the modulation amplitude to obtain the relevant limits for your paper.

3. L92, cite relevant literature for the expression for g_n . You may want to clarify how β_m is related to θ_m .

4. Fig. 2, please specify at which qubit frequency this is taken, and at that frequency, what is the detuning from the readout resonators. The detuning is important to understand at what modulation frequency the sidebands are expected to appear.

5. Fig. 2b, you seem to also observe a 1st-harmonic sideband around 600MHz. Is this a consequence of the finite detuning from the sweet spot? If so, why not trying to reproduce it in the theory plot of Fig. 2d by adding some finite detuning?

6. I find it somewhat confusing that the data in Fig. 2c is presented before those in Fig. 3a, on which they are based. In the text, you also refer to Fig. 3a before 2c. Why not moving panel c of Fig. 2 to Fig. 3?

7. You implicitly assume that the readout resonators are not thermally populated. Otherwise, I would expect the thermal occupation of the resonators to contribute to the thermal occupation of the qubits after applying the reset drive. Is this assumption justified? Based on your data, can you place a bound on the thermal occupation of the resonators? Or do you have an independent estimate for that?

8. To strengthen the scalability claim, further experiments could have been performed beyond the results of Fig. 4a. In particular: (a) simultaneous driving of two resets while a spectator qubit undergoes Ramsey, (b) repeated reset (to demonstrate that the protocol can be consistently reapplied without adding significant errors), and (c) full tomography on neighboring qubits, to exclude that the parametric modulation induces some other gate to occur between neighboring qubits (iSWAP, CZ...), which may not manifest itself in the experiments demonstrated in Fig. 4.

9. L155, when citing the original work on the Rabi population measurement technique (RPM), [10], you should also directly cite the follow-up work by MIT, [46], which introduced the more efficient pulse sequence that you are using in your experiment.

Referee 1

The authors demonstrate an unconditional qubit reset method, using a parametrically modulated flux pulse and a dissipative resonator. They use it on a 3-superconducting qubit device, demonstrating the reset of up to two qubits simultaneously, without significantly affecting the remaining one(s).

I have concerns about the analysis of the reset fidelity, which only assumes that the initial state is the first qubit excited state, and disregards the state of the resonator at the end of the reset pulse.

We thank the referee for the valuable comments, which has helped us greatly in improving our manuscript. Below we address the specific points raised:

Comment 1: The methods used to measure the ground (P_0) and first excited state population (P_1) [10 and 46 in the manuscript] both assume that higher level populations (P_2 or above) are negligible. From the estimated effective temperature of 75 mK, this is a reasonable assumption. However, it is widely reported that both two-qubit gates [Marques et al., arXiv:2102.13071] and readout [18] are notable sources of leakage to $|2\rangle$, which accumulates significantly throughout several error correction rounds. Therefore the claim that $|2\rangle$ is negligible cannot be substantiated beyond the steady (thermal) state.

Reply 1: *We agree that beyond the steady (thermal) state, leakage to the $|f\rangle$ state is not negligible. To address this, we extend our single-tone parametric modulation scheme to a two-tone scheme, specifically designed to achieve effective $|f\rangle$ state reset. Details are given in the new section “Reset of the $|f\rangle$ state: two-tone parametric reset” and Figure 4 in the main text. With the new protocol, we demonstrate that if the initial state is prepared as $|f\rangle(|e\rangle)$, the system will reach the ground state (readout floor) within 1000 (600) ns with a fidelity higher than 99.2% (limited by readout error).*

Comment 2: The protocol is only effective for initial state $|1\rangle$. The theory also addresses only the qubit subspace. Would it be possible to apply the same method to deplete $|2\rangle$, and subsequently deplete $|1\rangle$? If so, what time/fidelity would be expected?

Reply 2: *With the changes outlined in Reply 1, our new two-tone scheme also works to deplete $|f\rangle$. It is indeed possible to first deplete $|f\rangle$ and subsequently deplete $|e\rangle$, though we find it takes substantially longer time to reach the same fidelity than the scheme involving two-tone modulation simultaneously. Another benefit is that the two-tone scheme is unconditional on the initial states. The reset pulse is the*

same to deplete either $|f\rangle$ or $|e\rangle$, or any superposition of states in the $|g\rangle, |e\rangle, |f\rangle$ subspace. As mentioned above, with the new protocol, if the initial state is prepared as $|f\rangle(|e\rangle)$, the system will reach the ground state (readout floor) within 1000 (600) ns with a fidelity higher than 99.23%.

Comment 3: The highest speed of the reset pulse is measured by timing the reset pulse to the first minimum in the qubit $|e\rangle$ population. Although this is the ground state for the qubit, it is not the ground state for the readout resonator, as can be seen by the revival of qubit population over multiple exchanges with the cavity (Fig.2c) In order for any operation on the qubit to succeed with high fidelity or without dephasing induced by fluctuations in the photon number, the resonator must first be emptied. In the absence of active depletion of the resonator, one needs to wait $\sim 5/\kappa$ for the reset operation to be complete, similarly to the case when the flux pulse is left on throughout the whole time. If the authors motivate this technique in the context of repeated error correction (introduction), resetting both qubit and resonator before the next operation would take at least 800 ns (Fig.3b), which is not faster than other methods (as seen on Fig. S2). For the above reasons, I find the claim that this method “outperforms all existing reset schemes” (as stated in the conclusion and in Fig. S2) hardly justifiable. There is not enough information to quantify the ability to prevent or suppress leakage to higher excited states, and the reset time (for the intended application) does not appear to be any shorter than the cited methods.

Reply 3: We agree that the resonator must be emptied before error correction or if subsequent gates are immediately applied on the same qubit. However, the first minimum point has practical significance in the cases where there is no immediate gate on the same qubit or immediate gates are applied on other qubit, such as in state transfer (Nature 558, 264-267 (2018)) or quantum simulations (Nature 566, 51–57 (2019)). In the scenario where photon depletion of the resonator is a must, after the reset pulse is applied, a $\sim 5/\kappa$ passive wait duration (250ns) is needed to deplete the mean photon number below 0.01, corresponding to a negligible stark shift of 51.5 kHz. Taking the re-thermalization process (Fig. S5) into consideration, the net fidelity at the end of this process is 99.86%. An alternative is, as mentioned by the referee, active resonator reset. If speed is not a priority, one-tone reset with the modulation always on is an option, requiring 1000 ns and achieving 99.9% fidelity. When $|f\rangle$ Leakage is non-negligible, two-tone modulation is preferable and it takes 1000 (600) ns for both $|e\rangle$ and $|f\rangle$ ($|e\rangle$ only) to reach 0.77 % readout floor. In conclusion, the parametric reset schemes we proposed have a high degree of flexibility in different scenarios. We have summarized the performance and use cases of our protocols in Supplementary Table2. The $|f\rangle$ Leakage can be effectively depleted, as mentioned in **Reply 2**, by employing two-tone parametric modulation. We have updated Supplementary Fig.S2 to give a fairer comparison between our approach and previous proposals (as measured by both the time to the first

minimum, as well as the additional $\sim 5/\kappa$ time for resonator depletion), which now shows more clearly how our protocol outperforms those other schemes.

Comment 4: In the introduction “the syndrome qubits need to be reset on demand”. Technically, reset of the ancillas at every round of error correction round is not necessary (although definitely helpful in reducing leakage accumulation, see main comments above)

Reply 4: *We agree with the referee that “reset of the ancillas at every round of error correction round is not necessary”, However, here we mean the syndrome qubits need to be reset whenever required. We have revised the sentence to “the reset of the syndrome qubits needs to be accomplished with high fidelity in the time scale of a single qubit pulse.. ..”*

Comment 5: In the Supplementary Material:* “The population of the population” (beginning of note 4)

Reply 5: *Corrected as suggested.*

Comment 6: Fig. S2, some of the years are wrong: (Geerlings et al, 2013; Riste et al, 2012)

Reply 6: *Revised as suggested. Thank you for pointing out these errors.*

Comment 7: I find the distinction between active control and feedback control confusing, as feedback is a type of active control. Perhaps measurement and non-measurement based, or unconditional would be more clear.

Reply 7: *Revised as suggested.*

Referee 2

The authors describe and experimentally demonstrate a method for resetting a transmon superconducting qubit by means of a parametrically induced coupling to its readout resonator. Reliable active reset protocols such as this one are very important in the operation of superconducting quantum computing systems. The presented protocol is conceptually rather simple and I believe that many superconducting circuit labs have explored and tested similar approaches (swapping an excitation from the qubit to a resonator by parametric driving). However, to my knowledge, a clear exposition of such tests together with detailed quantitative analysis has so far been missing from the published literature.

The manuscript is well written, the explanation of the underlying principles clear and the experimental results well presented. I believe the scheme and its presentation would be highly relevant to researchers in the field of quantum computing with superconducting qubits and it warrants publication in Nature Communications with a few minor revisions.

We are pleased to read the referee's positive view of reliable active reset protocols such as ours. We address the specific points he/she raised below:

Comment 1: The authors refer to the recent work by McEwen et al. [18] as requiring careful calibration of flux pulses to achieve tuning of the qubit on resonance with the readout resonator. To my understanding, this is not exactly correct since the cited work tunes the qubit through resonance with the resonator to exchange the excitation adiabatically, which does not require careful adjustment of the frequency.

Reply 1: Thank you for pointing out this error. We have revised the sentence to "However, this process requires distortion correction of the flux control pulse.....".

Comment 2: Also, I am not sure if it is appropriate to represent the results of [18] in Fig. S2 by a square (active control at first minimum) rather than a circle (active control at 8τ). The 200-300ns shown in Fig. S2 by the data-point for [18] correspond to about 6 times the resonator relaxation time (45ns) and there are not really any minima to speak of since the protocol does not rely on resonant swapping. I think it would be fairer to show a circle for [18] (possibly showing a longer protocol time $8\tau = 360$ ns).

Reply 2: We thank the referee for pointing out there is no resonant swapping or “first minimum point” in the scheme in [18]. We have replaced the square with a circle as suggested. The data point corresponds to 250 ns and 10^{-2} as described in ref [18].

Comment 3: Since the authors claim quite prominently that their method outperforms all existing reset schemes, I think Fig. S2 which backs this claim (together with a short discussion) would deserve to be in the main text of the manuscript rather than in the supplementary material. I also believe it could be stated more explicitly that the first minimum points (indicated by squares in Fig. S2) represent the reset time only in settings where one does not need to reuse the protocol immediately. Such settings certainly exist and in these, the very fast excitation swap the authors demonstrate would have its use. However, in many other cases, one would still need to wait for the excitation in the resonator to decay before another reset can be performed with the same resonator. This sets an effective minimum time several times larger than $\sim 1/\kappa$. The authors correctly mention this in the discussion section but I think it is a rather important aspect that could be mentioned more prominently.

Reply 3: As suggested, we have discussed the first minimum points in detail in the Discussion section, and summarized the performance and use cases of our protocols in Supplementary Table 2.

Comment 4: Could the authors explain why the decay rate of the excited state population in curve B in Fig. 2c seems to be much slower than might be expected from the stated $1/\kappa_r$? If $1/\kappa_r$ is approximately 50 ns and therefore $1/\Gamma = 100$ ns, shouldn't we expect the population to decrease to approximately 0.37 in about 100 ns?

Reply 4: Indeed, the decay of the $|e\rangle$ population in curve B is slower than $1/\Gamma$. The actual decay behaviour of the excited state does not depend on the Γ only. As described in Supplementary Note 4, when the qubit is prepared at $|e0\rangle$, the time evolution of p_e can be derived as

$$p_e = P_{e|e}(t) = \begin{cases} e^{-\frac{\kappa_r t}{2}} \left(\frac{\kappa_r t}{4} + 1\right)^2 & |g_n| = \kappa_r/4 \\ e^{-\frac{\kappa_r t}{2}} \left[\cos(Mt) + \frac{\kappa_r}{4M} \sin(Mt)\right]^2 & |g_n| > \kappa_r/4, M = \frac{\sqrt{16|g_n|^2 - \kappa_r^2}}{4} \\ e^{-\frac{\kappa_r t}{2}} \left[\cosh(Mt) + \frac{\kappa_r}{4M} \sinh(Mt)\right]^2 & |g_n| < \kappa_r/4, M = \frac{\sqrt{\kappa_r^2 - 16|g_n|^2}}{4} \end{cases}$$

At the critical point when $|g_n| = \kappa_r/4$, p_e is $e^{-\frac{\kappa_r t}{2}} \left(\frac{\kappa_r t}{4} + 1\right)^2$ which has two terms, $e^{-\frac{\kappa_r t}{2}}$ is a dominant exponential decay term. The other square term $\left(\frac{\kappa_r t}{4} + 1\right)^2$ slows down the decay as in curve B in Fig. 2c. To avoid confusion, in the revised manuscript, we define Γ as the “reset rate” instead of the “decay

rate". We also added a discussion of the time-resolved population p_e in the Methods section, after the Γ expression.

Comment 5: Have the authors looked at how the protocol acts on initial states other than the excited state? Specifically, if the qubit is in the ground state to begin with, is there any chance that the protocol actually excites it? I would not expect this to happen in a parametric scheme (which should preserve the total number of excitations) but in my opinion it would be important to verify this explicitly. If the authors have investigated this, a brief sentence stating the highest observed excitation probability (or if this was unmeasurable, mentioning it explicitly) may be enough to address this.

Reply 5: When the system begins in $|g\rangle$, there is no experimental observation of the $|e\rangle$ or $|f\rangle$ populations. Based on our state readout fidelity, we conclude that the observed excitation probability is below 0.1%. As mentioned by the referee, the total number of excitations should be conserved and the reset frequency (hundreds of MHz) is much smaller than the qubit and resonator energy gap. We therefore expect the excitation probability to be negligible. We have added these estimates to Supplementary Note 5.

Comment 6: Have the authors analyzed the source of the background residual population? 2.4% (equivalent to 75mK) seems rather high and I wonder if this could be due to insufficient thermalization of some of the input lines. I would see this as a problem if this is caused by the rather low attenuation of the flux lines (Fig. S1 shows no attenuators in them), since I expect the rather strong parametric driving required for this protocol may not be compatible with higher attenuation. Can the authors comment on whether the effective temperature is limited by the flux line attenuation and whether that is in turn dictated by the required strength of the parametric drive?

Reply 6: The parametric drive (hundreds of MHz) was added through the z bias line with 30dB attenuation from the room temperature AWG to the sample (similar to other works as in [Barends, R. Nature 508,500–503 (2014)]). At each cold plate below 4K, the line was thermalized well by the attenuator with the heat sink. With the existing z bias configuration, we found it is sufficient to generate the parametric coupling required ($0.3 \Phi_0$ at most, similar value in Caldwell S. A. Phys. Rev. Applied 10, 034050 (2018)). In our case, both the parametric drive amplitude and frequency are limited by the source (AWG) we used. For "The rather low attenuation line", the referee may refer to the DC line. However, it only provides a static bias to the qubit. The 75mK effective temperature is comparable to

previous results (35mK-130mK) as mentioned in Jin, X. Y., et al PRL 114.24(2015), 240501, Riste, D. et al. PRL 109, 240502 (2012) and Johnson, J. E., et al. PRL 109.5 (2012): 050506. We attribute this to stray infrared radiation and insufficient thermalization of the sample box to the cold finger. We have added these possible causes to the main text (line179-line182), and agree that this issue warrants more careful investigation.

Comment 7: I am curious about the statement that "...residual population reaches 0.08%±0.08% at 34ns and remains below 0.2% after 800ns". Does this mean the population rises from the 0.08% achieved right after the reset pulse to 0.2%? If so, can the authors say what could be causing this? Is it just the spontaneous re-thermalization of the qubit with its effective 75 mK environment?

Reply 7: *The exact value after 500 ns in Figure 3b are as follows:*

Time(ns)	Residual
500	0.27 ± 0.17
1000	0.09 ± 0.19
1500	0.06 ± 0.15
2000	0.08 ± 0.14

After 1000ns, the residual population is below 0.1%, close to 0.08%, within the error bars. To be more precise, we have revised the sentence to "The minimum residual population reaches 0.08% at 34 ns and remains below 0.1% after 1000 ns".

Referee 3

The manuscript by Zhou et al. convincingly demonstrates the functionality described in the title, that is, a rapid and unconditional parametric reset protocol for frequency-tunable superconducting qubits. The protocol is based on parametric modulation of the frequency of one of two coupled modes with direct coupling and large detuning, a technique that has been extensively studied before, but not applied to qubit reset. Existing literature on the subject is well acknowledged and a critical comparison is drawn between present and previous realizations. Advantages of this reset protocol are its speed (the fastest to-date in the “timed mode”), absence of hardware overhead (the resonator used for the reset doubles as a readout resonator, same as in Magnard2018), and simplicity of operation. Possible disadvantages are that it only applies to frequency-tunable qubits, and possibly that – at least for the parameters explored here – a Purcell filter is needed to achieve coherence times larger than a few microseconds, due to the relatively large coupling strength and relatively small detuning between qubit and resonator.

The analysis of the reset protocol is scientifically sound. However, I was not convinced by the investigation of the effects of the reset protocol on neighboring qubits (see first comment below). The theory explanation of sideband driving could also be improved. In summary, the proposed scheme is a valuable addition to the existing reset schemes in the community, but I do not regard it as a major advance. For this reason, I am unsure whether this manuscript belongs to Nature Communications as opposed to a more specialized journal. In any case, I recommend that the authors address the comments below.

We thank the Referee for the valuable feedback and for pointing out the importance of the present work. The protocol also applies to fixed-frequency qubits, after slight modification. For example: if a tunable coupler with a readout resonator is added between the qubits (Youngkyu Sung, Phys. Rev. X 11, 021058), a parametric drive on the coupler will induce an interaction between the qubit and the coupler's resonator, which is similar to our proposed reset scheme.

Below we address the specific points he/she raised:

Comment 1: Of your three qubits, Q2 and Q3 have very bad dephasing times ($< 1\mu\text{s}$), while Q1 has a better one ($\sim 10\mu\text{s}$). Can you comment on possible reasons for this difference? In addition, when looking

at the effects of the reset protocol on neighboring qubits, it would make most sense to analyze the dephasing induced on your “best” qubit, Q1, while trying to reset Q2 (which you show you can do in Fig. 4b). Instead, you are only considering the qubits with very bad dephasing time and show that resetting the “good” qubit does not make it even worse. That does not seem like the best comparison you can make, and certainly does not justify your claim that the method “has negligible effect on neighboring qubits”, also given that state-of-the-art coherence times (T_2) for superconducting qubits are of the order of 100 μ s, not 1 μ s.

Reply 1: Q2 and Q3 have unsatisfying dephasing times mainly because they are away from the sweet spot (Applied Physics Reviews 6.2 (2019): 021318.). Especially Q2, at nearly 400 MHz away from the sweet spot. As suggested for the Ramsey-based experiments, we have also tried to analyze the Q2 dephasing while simultaneously resetting Q1 and Q3. Moreover, we have analyzed the Q1, Q3 Ramsey (since Q1 has the best dephasing property) while resetting Q2. The new data is summarized in Figure S6 of the revised version. We do not observe any degradation of the coherence or a frequency shift induced by the parametric drive on the neighbouring qubits.

Comment 2: L84, the statement “A sinusoidal modulation of the external flux...” is incorrect because of the nonlinear relationship between frequency and flux. In fact, in L128 you correct yourself with a statement that the non-specialist reader may not understand. Please follow the standard treatment of this process in which you start from the full expression and then take a first and second order expansion in the modulation amplitude to obtain the relevant limits for your paper.

Reply 2: Revised as suggested (line86-line95).

Comment 3: L92, cite relevant literature for the expression for g_n . You may want to clarify how β_m is related to θ_m .

Reply 3: Revised as suggested (line99-line101).

Comment 4: Fig. 2, please specify at which qubit frequency this is taken, and at that frequency, what is the detuning from the readout resonators. The detuning is important to understand at what modulation frequency the sidebands are expected to appear.

Reply 4: We have added the frequency information “ $\omega_q/2\pi = 5.784$ GHz $\omega_r/2\pi = 6.441$ GHz , $\Delta = \omega_q - \omega_r = -0.659$ GHz ” both in the main text and in the Fig.2 caption.

Comment 5: Fig. 2b, you seem to also observe a 1st-harmonic sideband around 600MHz. Is this a consequence of the finite detuning from the sweet spot? If so, why not trying to reproduce it in the theory plot of Fig. 2d by adding some finite detuning?

Reply 5: Yes, exactly. The 1st harmonic sideband around 600 MHz is caused by the small detuning (~ 1 MHz). We have reproduced it in Fig.2d as suggested. Thanks!

Comment 6: I find it somewhat confusing that the data in Fig. 2c is presented before those in Fig. 3a, on which they are based. In the text, you also refer to Fig. 3a before 2c. Why not moving panel c of Fig. 2 to Fig. 3?

Reply 6: We apologize for the confusion. Fig.2c is not based on the Rabi Population Measurement (RPM) in Fig.3a. The excited population P_e in Fig.2c is acquired by direct readout measurements. However, the exact reset fidelity is limited by measurement fidelity. We then performed the RPM and “two-point method” experiments in Fig.3. To conclude, Fig.2c shows the different features of the decay behaviour in each regime, corresponding to the A, B, C points in Fig.2b. We have added “Dots are the raw data acquired by direct readout measurements” in both the caption of Fig.2c and the main text to distinguish from the RPM method.

Comment 7: You implicitly assume that the readout resonators are not thermally populated. Otherwise, I would expect the thermal occupation of the resonators to contribute to the thermal occupation of the qubits after applying the reset drive, is this assumption justified? Based on your data, can you place a bound on the thermal occupation of the resonators? Or do you have an independent estimate for that?

Reply 7: Yes, we have an estimate of the thermal occupation of the resonator. When the average thermal population $\bar{n} \ll 1$, the thermal population induced dephasing is proportional to \bar{n} (Clerk, A. A. Phys. Rev. A 75, 042302(2007)):

$$\Gamma_\phi = \frac{1}{T_2^*} = \frac{\bar{n}\kappa\chi^2}{\chi^2 + \kappa^2}$$

where κ is the resonator linewidth and χ is the dispersive shift. From the measured value of T_2^* in Supplementary Table 1, and assuming thermal population is the only source of dephasing, we obtain $\bar{n} \leq 0.01$. This gives the upper bound of the mean photon number. Another estimate via the rate model gives a reasonable mean photon $\bar{n} \approx 0.01\%$ (details in Supplementary Note 6). In this model, we assume that the excitation rate of the resonator is the same as the qubit's due to the similar electromagnetic environment. As the decay rate of resonator κ_r is about 200 times larger than the qubit decay rate $1/T_1$, the mean photon number can be derived as 0.01%. Thus, thermal occupation of the resonators is indeed contributing to the population, but it is a negligible effect (corresponding to a negligible stark shift of 51.5 kHz for $\bar{n} = 0.01$). We have included this analysis in Supplementary Note 6.

Comment 8: To strengthen the scalability claim, further experiments could have been performed beyond the results of Fig. 4a. In particular: (a) simultaneous driving of two resets while a spectator qubit undergoes Ramsey, (b) repeated reset (to demonstrate that the protocol can be consistently reapplied without adding significant errors), and (c) full tomography on neighboring qubits, to exclude that the parametric modulation induces some other gate to occur between neighboring qubits (iSWAP, CZ...), which may not manifest itself in the experiments demonstrated in Fig. 4.?

Reply 8: We thank the referee for these valuable suggestions.

(a): We have performed the suggested experiment and summarized the results in Figure S6.

(b): As suggested, after the reset finishes, we apply another pi pulse to excite and reset again, repeated 100 times with both one-tone and two-tone parametric reset, and no apparent degradation of the reset fidelity or excitation is observed (Figure S8).

(c): For the gate fidelity, we have performed Clifford Randomized Benchmarking measurements on the adjacent (Q2) and the next-nearest qubit (Q3) to see if the reset pulse degrades the gate fidelity. We find that the average single gate fidelity variation between having the reset on and off is 0.08% (0.03%) for Q2(Q3) with one-tone parametric reset and 0.06% (0.01%) for Q2 (Q3) with two-tone parametric reset. We have summarized the results in Figure 4 and Figure S9.

Comment 9: L155, when citing the original work on the Rabi population measurement technique (RPM), [10], you should also directly cite the follow-up work by MIT, [46], which introduced the more efficient

pulse sequence that you are using in your experiment.

Reply 9: Reference added as suggested.

Reviewers' Comments:

Reviewer #1:

Remarks to the Author:

In this revision, the authors have expanded their manuscript to include:

* Reset from the second excited state $|f\rangle$. They added new data (Fig. 4) and a new section to introduce a two-tone reset protocol that achieves unconditional reset from any state in the subspace comprising the first three transmon levels. They included theory (Supplementary Note 8), data, and simulation of this modified protocol.

* A better distinction between qubit reset and joint qubit plus resonator reset, with an explanation on cases where one or the other are relevant (Discussion). They have also included a Supplementary Note 3 and Table S2 to compare different variants of their protocol.

* All of my other comments were also addressed.

Because of the above, I believe the authors have amply responded to my concerns and significantly improved their manuscript as a result. I can therefore recommend publication to Nat. Commun. without reservation.

Reviewer #2:

Remarks to the Author:

I would like to thank the authors for taking a considerable amount of effort to address my comments as well as those of the other referees. The questions I had were answered generally to my satisfaction and I recommend publication in Nature Communications.

I have one remaining remark: I would still urge caution about the claims of the need for distortion correction in the work of McEwen et al. (lines 47-48) The protocol used there is (quasi)adiabatic and so should be rather resistant to pulse distortions. I cannot find any mention of pulse distortion correction in McEwen's paper. Otherwise, I agree with the authors' statement that in "dc-tuning" approaches such as that of McEwen, cross-talk to neighboring qubits may cause issues (which would presumably be mitigated by using parametric coupling as in this work).

Reviewer #3:

Remarks to the Author:

The authors have satisfactorily replied to all referee's comments. To address some of the comments, they have added a sizable amount of new data and analysis to the manuscript, most notably, the implementation of a two-tone reset scheme to reset the $|f\rangle$ state, and Clifford randomized benchmarking of neighbor qubits while the target qubit was being reset.

Taking all referees' remarks together, a picture emerges in which the advantage offered by this reset technique appears less decisive and restricted to some applications. Waiting for the readout resonator to ring down would take almost 300ns, much larger than the 34 ns for the case in which only e-state is populated and the emptying of the resonator discarded. If the $|f\rangle$ level needs to be emptied, the reset process takes place over a time of 1000ns, more than a factor 2 larger than the result of Magnard2018.

In summary, it is my opinion that the results presented, while certainly interesting and worth publishing, would be a better fit to a more specialized journal, for example, npj Quantum Information. In addition, please consider the technical comments below.

1) Fig. 4a, caption "prepared in the $|f\rangle$ ", add "state".

2) I do not understand Figure S6. Q1 has a T_2^* of 11 μ s, Q2 and Q3 around 1 μ s (according to Table S1). So how is it possible that, say, in panel (d), the Ramsey fringes of Q1 and Q3 decay with the same time constant?

3) Suppl note S6, "We assume that the excitation rate of the resonator from $|0\rangle$ to $|1\rangle$ is the same as the qubit's excitation rate from $|g\rangle$ to $|e\rangle$, as they experience similar electromagnetic environments."

This assumption can be challenged. The readout resonator is a low-Q mode overcoupled to a coplanar waveguide, an Ohmic environment thermalized by a cold resistor in the attenuation chain. Its steady-state population is dictated by the temperature of the waveguide. The qubit, by contrast, is undercoupled to measurement and control wiring and thermalizes to its own bath, most likely a collection of two-level fluctuations at various interfaces. We know little about what drives this bath out of equilibrium. Generally, people have found it is more difficult to thermalize undercoupled modes. In fact, you have 2% thermal occupation in the qubit, but a much smaller occupation in the resonator. It is incorrect to say that this is because the decay rate of the resonator is faster. It is because the bath to which the resonator is coupled is colder.

Reviewer #1 (Remarks to the Author):

In this revision, the authors have expanded their manuscript to include:

* Reset from the second excited state $|f\rangle$. They added new data (Fig. 4) and a new section to introduce a two-tone reset protocol that achieves unconditional reset from any state in the subspace comprising the first three transmon levels. They included theory (Supplementary Note 8), data, and simulation of this modified protocol.

* A better distinction between qubit reset and joint qubit plus resonator reset, with an explanation on cases where one or the other are relevant (Discussion). They have also included a Supplementary Note 3 and Table S2 to compare different variants of their protocol.

* All of my other comments were also addressed.

Because of the above, I believe the authors have amply responded to my concerns and significantly improved their manuscript as a result. I can therefore recommend publication to Nat. Commun. without reservation.

We thank the referee for the positive comments and recommendation.

Reviewer #2 (Remarks to the Author):

I would like to thank the authors for taking a considerable amount of effort to address my comments as well as those of the other referees. The questions I had were answered generally to my satisfaction and I recommend publication in Nature Communications.

We thank the referee for the positive comments and recommendation. We address the specific point he/she raised below:

Comment 1: I have one remaining remark: I would still urge caution about the claims of the need for distortion correction in the work of McEwen et al. (lines 47-48) The protocol used there is (quasi)adiabatic and so should be rather resistant to pulse distortions. I cannot find any mention of pulse distortion correction in McEwen's paper. Otherwise, I agree with the authors' statement that in "dc-tuning" approaches such as that of McEwen, cross-talk to neighboring qubits may cause issues (which would presumably be mitigated by using parametric coupling as in this work).

Reply 1: *We agree with the referee that the distortion correction is not mentioned in McEwen et al. We have followed the suggestion and revised the sentence to "For instance, in one such approach, flux control is used to tune the qubit frequency to that of the resonator rapidly. However, this process significantly affects neighbouring qubits via crosstalk."*

Reviewer #3 (Remarks to the Author):

The authors have satisfactorily replied to all referee's comments. To address some of the comments, they have added a sizable amount of new data and analysis to the manuscript, most notably, the implementation of a two-tone reset scheme to reset the $|f\rangle$ state, and Clifford randomized benchmarking of neighbor qubits while the target qubit was being reset.

Taking all referees' remarks together, a picture emerges in which the advantage offered by this reset technique appears less decisive and restricted to some applications. Waiting for the readout resonator to ring down would take almost 300ns, much larger than the 34 ns for the case in which only e-state is populated and the emptying of the resonator discarded. If the $|f\rangle$ level needs to be emptied, the reset process takes place over a time of 1000ns, more than a factor 2 larger than the result of Magnard2018.

In summary, it is my opinion that the results presented, while certainly interesting and worth publishing, would be a better fit to a more specialized journal, for example, npj Quantum Information. In addition, please consider the technical comments below.

We thank the referee for the positive comments. We address the specific points he/she raised below:

Comment 1: *Fig. 4a, caption "prepared in the $|f\rangle$ ", add "state".*

Reply 1: *Revised as suggested.*

Comment 2: *I do not understand Figure S6. Q1 has a $T2^*$ of 11us, Q2 and Q3 around 1us (according to Table S1). So how is it possible that, say, in panel (d), the Ramsey fringes of Q1 and Q3 decay with the same time constant?*

Reply 2: *We thank the referee for pointing out this. When we performed the complementary experiments after receiving the referee's first comments, the parameters of three qubits changed due to our dilution refrigerator's abnormal cooling cycle (we have included this information in Supplementary Table 1).*

The Ramsey fringes of Q1, Q3 decays much faster than expected. One possible cause is the beating pattern in these two fringes. As shown in the figure below, due to the beating pattern, the fitted exponential decay time constant between 0 and 1000 ns (orange curve, similar to those in Figure S6 d) is much shorter than the one between 0 and 10 us (green curve). We have also included the fitting of the Ramsey fringes with a beating pattern in Supplementary Note 11.

Figure: *Q1 (left) and Q3 (right) 's Ramsey fringes with beating pattern.*

Comment 3: "We assume that the excitation rate of the resonator from $|0\rangle$ to $|1\rangle$ is the same as the qubit's excitation rate from $|g\rangle$ to $|e\rangle$, as they experience similar electromagnetic environments."

This assumption can be challenged. The readout resonator is a low-Q mode overcoupled to a coplanar waveguide, an Ohmic environment thermalized by a cold resistor in the attenuation chain. Its steady-state population is dictated by the temperature of the waveguide. The qubit, by contrast, is undercoupled to measurement and control wiring and thermalizes to its own bath, most likely a collection of two-level fluctuations at various interfaces. We know little about what drives this bath out of equilibrium. Generally, people have found it is more difficult to thermalize undercoupled modes. In fact, you have 2% thermal occupation in the qubit, but a much smaller occupation in the resonator. It is incorrect to say that this is because the decay rate of the resonator is faster. It is because the bath to which the resonator is coupled is colder.

Reply 3: *We agree with the referee that the bath of the resonator and the qubit is different, especially quasiparticle will thermalize the qubit seriously. We have deleted the estimation under this assumption. The upper bound estimation will be only given by thermal population induced dephasing as in Supplementary Note 6.*